# Multinomial Naive Bayesian Classifier Framework for Systematic Analysis of Smart IoT Devices

**DOI:** 10.3390/s22197318

**Published:** 2022-09-27

**Authors:** Keshav Kaushik, Akashdeep Bhardwaj, Susheela Dahiya, Mashael S. Maashi, Moteeb Al Moteri, Mohammed Aljebreen, Salil Bharany

**Affiliations:** 1School of Computer Science, University of Petroleum and Energy Studies, Dehradun 248007, India; 2Software Engineering Department, College of Computer and Information Sciences, King Saud University, Riyadh 11451, Saudi Arabia; 3Department of Management Information System, College of Business Administration, King Saud University, P.O. Box 28095, Riyadh 11437, Saudi Arabia; 4Department of Computer Science, Community College, King Saud University, P.O. Box 28095, Riyadh 11437, Saudi Arabia; 5Department of Computer Engineering & Technology, Guru Nanak Dev University, Amritsar 143005, India

**Keywords:** artificial intelligence, alexa, amazon, internet of things, machine learning, natural language processing, smart devices

## Abstract

Businesses need to use sentiment analysis, powered by artificial intelligence and machine learning to forecast accurately whether or not consumers are satisfied with their offerings. This paper uses a deep learning model to analyze thousands of reviews of Amazon Alexa to predict customer sentiment. The proposed model can be directly applied to any company with an online presence to detect customer sentiment from their reviews automatically. This research aims to present a suitable method for analyzing the users’ reviews of Amazon Echo and categorizing them into positive or negative thoughts. A dataset containing reviews of 3150 users has been used in this research work. Initially, a word cloud of positive and negative reviews was plotted, which gave a lot of insight from the text data. After that, a deep learning model using a multinomial naive Bayesian classifier was built and trained using 80% of the dataset. Then the remaining 20% of the dataset was used to test the model. The proposed model gives 93% accuracy. The proposed model has also been compared with four models used in the same domain, outperforming three.

## 1. Introductions

Intelligent voice assistants, such as Microsoft’s Cortana, Apple’s Siri, Amazon’s Alexa, and Google’s Assistant, have grown significantly in popularity and use over the past several years. These intelligent voice assistants have changed how users interact with smartphones or computers. Individuals are using these intelligent voice assistants to give voice commands and get the appropriate information like daily news, weather reports, or fulfilling orders like playing media. Along with these uses, voice assistants are also used to perform basic tasks like setting timers or alarms and making phone calls. Nowadays, these voice assistants, especially Alexa, are also used in intelligent IoT-enabled devices to support voice control. The voice assistants are connected to the internet. Whenever a user gives any voice command, that command is sent to a central computing system for analysis. In the central computing system, the voice assistants analyze and translate the knowledge using natural language processing (NLP), and the voice assistant provides a proper response for that command. Recent advances in NLP have allowed voice assistants to generate meaningful responses rapidly [1].

With the help of artificial intelligence (AI), intelligent voice assistants can also detect or understand the user’s emotions and perform sentiment analysis. Sentiment analysis plays an important role when users share their feedback or experience regarding some product through voice assistants. Using sentiment analysis of voice assistants, commercial business companies can use insights to improve their products or services. It is essential for the voice assistant to accurately detect the sentiment in the users’ feedback or product review and analyze it to detect the user’s correct tone and mood. With the help of AI and NLP, the magnitude of the mood and tone of a user can be calculated, and a numerical score can be assigned to them. Depending on the outcome of the sentiment analysis, proper assistance can be provided to the user. As the popularity of intelligent voice assistants is increasing, the number of supported services is also growing rapidly. After using a product, most users like to share their experience about the product by writing reviews. This review not only helps the potential buyers but also helps the business companies make sound, impactful decisions about the product. So, it is essential to perform sentiment analysis on users’ reviews about the product and the services provided by that product. The product reviews are primarily available in text format. AI-based sentiment analysis can classify the products’ reviews into positive or negative categories by looking at the words used in the study. Generally, a positive product review contains words like good, easy, love, happy, and great, and a negative review contains words like disappointing, challenging, frustrating, harmful, waste, not, and annoy.

The authors have predicted customer sentiment from genuine Amazon Echo customer reviews using NLP in this research. The main objective of this research work is to indicate whether the users are happy or not with the Amazon Echo. If the customers are not satisfied with the product, then amazon can figure out the reason and help the users with proper assistance and update the product based on the reviews. In this research, a machine learning model has been built and trained to analyze thousands of studies of Amazon Echo to predict customer sentiments.

The remainder of this work is organized as follows:The Related Work section highlights the relevant research done in the same area.The Research Methodology section describes the step-by-step implementation of the method used in this research.The Results and Comparison section compares the results with similar models in the same domain.Conclusion sections conclude the investigation.

## 2. Related Work

The authors researched 354 journal publications since 2018 from IEEE, Springer, MDPI, ACM, Elsevier, and other highly referred journals. The papers were classified based on the research reviews, keywords, and results to match this research’s relevant work and results. The authors shortlisted relevant research as per the below selection methodology to finalize 35 investigations, as illustrated in Figure 1.

The final 35 research selected were categorized as illustrated in Table 1 for works closely matching the metadata, summary, keywords Machine Learning, Artificial Intelligence, Voice Assistant, and IoT devices, among others. The classification provided an overall distribution ratio between 17 to 24 percent.

Even though voice assistants are becoming more popular, a user-centric usability study is essential to assure their success. The System Usability Scale is a standard usability tool in a Graphical User Interface (GUI) setting. However, since there are substantial differences between GUI and speech-based systems, SUS’s relevance in a voice setting is uncertain. The current project has two objectives: to see whether SUS is acceptable for accessing the applications of voice assistants and to create a qualitative scale based on SUS that implements the specific aspects of speech-based communication. A group of researchers developed the Voice Usability Scale [2]. To achieve the objectives, a subjective test involving 62 persons is conducted. According to exploratory factor analysis, SUS has several issues when analyzing voice usability. In addition, the most acceptable VUS factor structure emphasizes three essential components: usability, emotiveness, uniqueness, and exposure. The data presented here should serve as a helpful starting point for creating a sound theoretical and practical basis for subjectively voice-based numeric rating assessment.

Among other voice assistants, Siri, Alexa, and Google Assistant have lately been the subject of intense debates over artificial intelligence, surveillance, gender roles, and security [3]. Nevertheless, the idea that virtual assistants are web platforms with the potential to affect how users to access, interpret and utilize the internet has received less attention. Ref. [4] focused their efforts in this area by emphasizing several problems that might lead to more observations and discussions in media studies, communication, and other related fields. This article examines the role of web platforms in regulating information access, the relationship between online manufacturing and utilization, and the significance of emotion in directing engagement with web resources. Investigating these issues in the context of intelligent speakers not only aids in placing these advances within broader discussions of communication and media studies. However, it also speaks to the fact that virtual assistants elevate exciting questions for online research and intellectually stimulating hypotheses about what the internet looks like, as speech has become one of the superior abilities to obtain information and resources online.

Ref. [5] aimed to provide the most up-to-date research on natural language production using deep learning techniques. The authors cited current research from renowned natural language processing experts and demonstrated how it might be used in Chatbot systems. The study began with a review of background information on deep learning for NLP, followed by a discussion of approaches ranging from the most basic to the most advanced. The authors also demonstrated the methods employed in systems like Apple’s Siri, Microsoft’s Cortana, Google Assistant, Amazon Alexa, and Facebook’s M and Chatbot systems, which all use spoken dialogue systems.

Representation learning is a complex problem to solve in various applications, and it is essential for successfully fulfilling regression tasks. The authors [6] developed a special framework for the Spoken Language Processing system that teaches interconnected domain representations, intentions, and slots by hierarchically utilising their interdependence. By pooling the portrayals of the places, the system learns the predictive model of choices based on the spaces linked to these intents. Integrating the representations of the domain’s intentions teaches the vector representation of the environment. This was the first approach for learning the hierarchical links between part, purpose, and slot representations. As shown by increased efficiency on the contextual cross-domain re-ranking task, the experimental findings demonstrate the efficacy of the drawings learned by our model. Artificial Intelligence (AI) is a type of artificial intelligence used to solve. Assistants are becoming more common in both homes and businesses. They have already departed their initial homes of “intelligent speakers” with simple access to music libraries. Ref. [7] looked at intelligent digital assistants to establish the groundwork for a complete technique. They have created a slew of new gadgets and are already filling things like television sets. Developing platforms around the fundamental functions of intelligent digital assistants is typical. The assistants’ AIS capabilities provide new services and build new business process interfaces. There are beneficial network effects between the helpers and the benefits inside the services. Consequently, many companies see the need to become engaged in the area of digital assistants, but few have a structure in place to connect their efforts to their broader company plan. The authors designed a strategic challenges and opportunities methodology based on their results.

Medicine, voice recognition, Siri Alexa, and machine learning are just a few fields where Deep Learning is gaining ground. Although machine learning is a trendy area in data science, every one of these technologies relies on Big Data Analytics. Big data is a compilation of massive amounts of data and information obtained from various sources, including websites, social networking sites, and other networks. In everyday life, it is used. When this data is gathered from multiple sources, such as Facebook, Whatsapp, Twitter, and others, it may be used to generate different results, including privacy guidelines, investment, stock markets, business, and research processes, to mention a few. Due to the lack of hidden layer interaction, the critical gap in this forecast was to search the online sources and the real-time segment of the population for its accuracy in share market fluctuations. Ref. [8] proposed sentiment classification, a deep learning accuracy prediction method that solves the problems and poor performance of doc2vec, long short term memory, and acts as an outstanding technology demonstration for our emotional model to determine the best stock market evaluation by using bi-directional long-short term memory and deep belief networks.

Deep neural networks are presently state of the art for numerous modern artificial intelligence applications. One of its effects has most likely influenced you. Facebook’s automated contact tagging, personal digital assistants, and Google Photos’ autonomous album building are all examples of DNNs in operation. Even though DNNs were originally employed in picture recognition, NLP, Translation Software, Spam and Intrusion detection, and Speech Recognition, the number of applications that utilize them has increased considerably. A smart grid is a system that provides a bi-directional transfer of information and energy in both directions. The energy flow is bidirectional since distributed generation and conservation may be connected to the grid through a mechanism. Data flows from intelligent devices to the central server, and control signals travel from the domain controller to the intelligent devices in a bidirectional information flow. Ref. [9] looked at where Deep Learning and Artificial Neural Networks are now, as well as some of the industry’s potential in Smart Metering and Smart Grid. Virtual assistants such as Bixby, Google Assistant, Cortana, and Alexa simplify life by inferring intent from the user’s input utterance. Virtual assistant adoption is more challenging in multilingual nations, such as India than in monolingual ones. People in multilingual societies regularly blend linguistic elements from many languages in a single remark. This process leads to code-mixing, which lacks traditional semantic and syntactic language traits compared to monolingual input. NLP is tricky when it comes to deciphering intended meaning from code-mixed data. There is a lack of applicable corpora and research for intent assessment on code-mixed text. The first Hindi-English Code-Mixed dataset for Intent Classification is presented in this paper and includes the top 10 features of a Virtual Assistant. For inferring intent from code-mixed text, ref. [10] proposed a deep learning-based approach. We compare the appropriateness and performance of several state-of-the-art techniques in an empirical analysis. On our dataset, our method achieves 96.68 percent accuracy.

Chatbots are artificial intelligence programs that converse with clients in natural language. The researchers sought to investigate whether a chatbot could fool customers into believing they were talking with real humans. Natural Language Processing may be done with the help of NLTK, a Python module. It is used to analyze speech input and generate responses similar to those given by individuals. Speech-based search engines, as well as virtual agents like Siri, Cortana, Google Assistant, and Alexa, are in great demand these days. Chatbots are in significant need right now, especially in the business sector, for simplifying customer support and reducing human labor costs. In conversation systems, a chatbot is generally used to gather information. Ref. [11] examined the various Chatbot technologies and the design and execution of a Chatbot system.

An intelligent chatbot is a computer program that can converse with humans and respond to queries on a particular subject. The current challenge is to create a system that resembles a human brain. In contrast to a centralized computer file system, the brain retains memories in a decentralized way across the brain with the aid of neurons. Short-term and long-term memory storage are available, each with a varied priority, dependent on the scenario. The system may accept written or spoken inputs and answer the query using a knowledge base. Chatbot has no problem-solving ability. Ref. [12] used deep neural networks to tackle data structure issues. The system can provide services to access data in formats such as arrays, stacks, queues, and trees with a specified dataset. The approach handled issues including traversing lists, reversing numbers, and interpreting syntactic divergences’ language using these data structures. The learning service is a DNN-trained model rather than an algorithmic program. The development of a problem-solving Chatbot will allow it to learn how to arrange and retrieve data based on the data structure chosen by the user. The authors employed a Neural Stack Machine with a Recurrent Neural Network as a controller.

Only ten percent of an iceberg is visible because ninety percent of it is considered hidden under the surface, in the water’s hidden depths. According to [13], many human emotion signals fit this property and may be found in interpretative data from interactions with standard IoT devices. In the form of face analysis, NLP, and voice analysis, the visible ‘tip of the iceberg’ comprises the most often studied “tells” of emotion. These provide a one-of-a-kind frozen snapshot of a person’s emotional situation. IoT has ushered in a cultural change in which sensors and applications collect data on many aspects of daily life. Hub devices, such as Voice Command Devices, gather and organize this information, which Smart Assistants like the Amazon Echo and Alexa may then access. Emoto-graphic Modelling is a ground-breaking idea that demonstrates how a variety of digital indicators accessible via these hubs may be used to assess a person’s emotional state. The resulting ‘Emoto-graphic’ classifications are based on statistical data collected from digital emotion signals. IoT, Cloud, and ML may be used to probe the iceberg’s inferential depths, delivering data on sleep, nutrition, exercise, and other habits. Emoto-graphic Iceberg’s complex “hidden” component may reveal patterns that indicate emotion over time. Modifications in these tendencies might help physicians, and marketers better understand a person’s mental health. The authors provide preliminary findings that show how a sequence of inquiries asked of an Amazon Echo Voice-Activated Device linked to the IoT might lead to a sense of melancholy. Machine learning algorithms that understand human-to-human contact might produce a more natural consumer experience for communal devices like Amazon Alexa. Among other indications such as vocal activity and gaze, a person’s audio-visual expression mainly blends voice tone and facial expression, and functions as an implicit indication of engagement between parties in a debate. Ref. [14] looked at deep-learning approaches for audio-visual identification of user expression. On the Aff-Wild2 database, the proposed strategies outperform the basic architecture with recurrent layers, with absolute increases of about 2% for arousal and polarity descriptors. In addition, multimodal designs outperform single-modality models by up to 3.6 percent. Ablation analyses in the Aff-Wild2 database emphasize the relevance of the visual medium for expression determination.

This study aimed to develop and install AI-based Voice Recognition System Software, as well as a package that included the integration of numerous tasks that we could process and execute using individual customers’ voice commands. AI-Based Virtual or Personal Assistant (IVA or IPA) is an acronym for AI Technologies Based Virtual or Personal Assistant. It is a collection of intelligent mechanisms that conduct various activities and services in response to queries and orders to operate intelligent systems. It is recognized as a square measure in certain areas of fields of study used for voice identification, in which voice recognition is identified as a channel of communication among individuals who will be alerted by machine, in which devices can recognize a person’s voice where the voice is the only method of double interaction. It enables to free-up both hands and vision doubtless for execution. The authors focused on AI systems that aid voice recognition and NLP. This module’s most essential features are NLP, speech recognition, and speech management. The voice assistant is one of these tools, and it can be implemented into many intelligent systems. Using your voice rather than a keyboard might help you save a lot of money. Ref. [15] put various speech recognition models, techniques, and approaches to the test, including those employed by other researchers in the field. Our basic configuration integrates as many tasks as feasible and executes them by voice command. The OpenCV method’s design and implementation are discussed, followed by experimental analysis and a graphical depiction of the findings.

IoT in the healthcare sector [16] is having an impact on people’s lives. The use of cutting-edge wearable technology simplifies life for individuals afflicted with various diseases. Several wearable technology products have features that make them suitable for integrating the human body. The newest technologies, such as edge computing, are helping wearables perform as planned. In this study [17], the authors provided a systematic way to comprehend IoT security needs, which will aid in the future design of safe IoT systems. The authors presented various situations and discussed potential IoT dangers and assaults while establishing these needs. The authors categorized potential risks and assaults into five categories based on the features of the IoT: telecommunications, hardware, people, mobility, and resource convergence. This study’s [18] main contribution is the classification of machine learning techniques, which explains how various methods are used to analyze the data in the process of extracting higher-level data. We’ll also talk about the possibilities and difficulties of machine learning for IoT data analytics. A use case of deploying an SVM to traffic data from the Aarhus smart city is provided for a more thorough investigation.

Since a significant quantity of data needs to be handled intelligently, a machine learning technique is essential for effectively deploying the IoT-powered wireless sensor networks (WSN) for this purpose. The use of AI-powered IoT and WSNs in the healthcare industry will be thoroughly covered in this article. For further research projects [19], this study will serve as a foundation for understanding the mechanism of the IoT in smart cities, particularly in the healthcare industry. This study [20] uses the phrase “smart transportation” to refer to a broad category that includes applications for infrastructure, parking, street lighting, accident identification and prevention, and route planning. To gain a comprehensive understanding of the developments in the aforementioned sectors and identify potential coverage requirements, this study aims to conduct a self-contained assessment of ML approaches and IoT applications in Intelligent Transportation Systems. It is clear from the examined publications that there may not be enough machine learning (ML) expertise for intelligent lighting and innovative parking technologies.

Due to recent, fast development, IoT technology allows many heterogeneous systems to connect. IoT has a reasonably sophisticated design due to the system’s connection with a wide range of devices and services. An overview of the urban IoT system, which is intended to enable smart cities and cutting-edge modern communications, is provided in this article [21,22,23,24,25,26,27,28].

## 3. Research Methodology

In this paper, the authors have used Natural Language Processing techniques (NLP) to predict customer sentiment from genuine amazon Echo customer reviews. The dataset used for the implementation is https://www.kaggle.com/sid321axn/amazon-alexa-reviews/kernels (accessed on 15 July 2022). Amazon Echo Alexa is a type of virtual assistant. It can be placed anywhere in the home, and you can ask Alexa anything you want if you want it, for example, to book an appointment if you’re going to make a phone call, if you want to play music, you can say hey, Alexa do something. The main objective of this research is to try to predict customer sentiment through amazon reviews for the echo product. This research helps know whether the customers are happy or not with the product. Suppose the customers are not satisfied with the product. In that case, the manufacturer should know the reason behind customer dissatisfaction so that the organization can update the product based on the reviews. To analyze thousands of customer evaluations to forecast whether or not consumers are satisfied. This procedure might be carried out automatically with no involvement from humans. For analyzing a Smart device like Amazon Alexa, artificial intelligence is helpful. The authors have built a machine learning model to explore the IoT-based smart device.

Figure 2 illustrates the research methodology that the authors have followed in the implementation. The entire process is divided into four significant steps. The authors have implemented it in the python programming language. Initially, all the necessary and required libraries related to the implementation are imported, and exploratory data analysis is performed. In the second step, the data visualization of amazon Alexa reviews is performed, and word cloud is implemented. After that, the data cleaning and tokenization are done to improve the dataset quality. In this step, the dataset is prepared and ready for model development. In the final step, a machine learning model to analyze an IoT-based Smart Device is built and trained.
Step 1: Import the required libraries and dataset to perform exploratory data analysis

In the first step, the required libraries like NumPy, Pandas, Matplotlib, and Seaborn are imported for implementation. Seaborn is used for effective data visualization by plotting heatmap. Pandas are used to manipulate the data and to perform exploratory data analysis. The dataset contains the amazon Alexa reviews given by various users. The information related to the dataset is shown in Table 2 given below.

As you can see in the table above that there are five columns in the dataset, out of the two columns rating and feedback are of integer type. Whereas, data, variation, and verified_reviews are of the object type. The rating column shows how many stars the customer had given to the product like a 1–5 rating. The date denotes the date on which the review is given. The variation shows the type of variant of Amazon Echo as if we have a Black Dot model, Charcoal Fabric model, Walnut Finish model, etc. The verified reviews are the actual reviews that we care about for implementation. That is the actual text that the authors have analyzed. Then finally, the last column is the feedback which is the sentiment. This is simply the customer sentiment, which is either zero or one. If some users say that–“I love my echo!” then that is positive sentiment and is represented by one whereas Zero represents the negative sentiment. Therefore, the rating and feedback column is important for visualization and training the machine learning model. As you can see in Table 3, there are 3150 rows in the dataset that has a non-null count. The statistical summary of the dataset is given in Table 3.

As you can see in Table 3, the average rating is around 4.46 approximately the standard deviation is 1.06, which is the dispersion away from the mean is one approximately. The minimum review of the course is one star. The maximum review of the course is five stars. The statistical summary has 25%, 50%, and 75% as well. On average, the average ratings of around 3000 reviews. It stands around 4.46 approx. The statistical summary gives the overall mathematical view of the dataset for analyzing it.
Step 2: Perform the data visualization and plot the word cloud for amazon Alexa reviews

In this step, the data visualization is performed and the word cloud of amazon Alexa reviews is plotted to check positive and negative comments. As the authors have highlighted in the previous step, for building the deep neural network, we have focussed on two columns namely–rating and feedback. Therefore, the data visualization for these two columns specifically is important. Figure 3a represents the rating counts, as you can see that more than 2500 users out of 3150 users have given a five-star rating to the amazon Alexa product. Around 500 customers have given four stars rating and very few customers who are not happy gave it one star or two stars.

Figure 3b shows the plot of feedback counts, around 2800 customers are happy with the product and they have given positive feedback i.e., represented by one, and approximately 250 customers out of 3150 feedbacks are not happy with the product and they have given negative feedback represented by zero. As per the feedback plot, there are more satisfied customers as compared to unsatisfied customers.

From the data visualization, the authors have observed that there are zero missing or novel elements in the dataset. Moreover, the customers are also happy with the product as many customers gave it five-star reviews and many customers give it the sentiment of one indicating that they are happy with the product. After that, the authors performed some basic data exploration as shown in Figure 4, and calculated the length of the verified_review column. After observing the review’s length plot, we can conclude that there are a lot of reviews that are short in length. Maybe some customers are highly satisfied with the product and they wrote 2–3 words reviews and a few customers who are unhappy with the product, they wrote lengthy reviews.

After plotting the histogram of the length of the review, the authors have plotted the word cloud of positive and negative reviews. Word cloud is a really powerful visualization that will enable us to gain a lot of insight from text data. Here you can observe in Figure 5a, the wordcloud of positive reviews that were plotted by joining all positive reviews into one large string. All the sentences are clubbed together to form a large string and then the wordcloud was plotted.

Here you can observe in Figure 6, the wordcloud of negative reviews that were plotted by joining all the negative reviews into one large string. All the sentences are clubbed together to form a large string and then the wordcloud was plotted.

The Algorithm 1 used for the creation of wordcloud of positive and negative reviews is given below:
**Algorithm 1** Wordcloud GenerationStep 1: Convert all the positive/negative verified reviews into separate sentencesStep 2: Calculate the length of all those sentencesStep 3: Join all the verified positive/negative into a single large stringStep 4: Print that generated single large stringStep 5: Import the WordCloud python package and plot the positive/negative wordcloud of that joined single string
Step 3: Perform data cleaning and tokenization

In step 3, the data cleaning and tokenization are performed. Before building up the model, the data is cleaned first. Cleaning the data means that several stopwords, punctuation marks, exclamation marks, and common words are removed. The idea of cleaning the data is to convert text data into just a bunch of numbers. Once we have converted the text data into numbers we can use that data to train a machine learning model using these numbers to perform sentiment analysis. The pipeline is created to clean all the messages. Message cleaning means removing punctuation, and stopwords. The Algorithm 2 used to perform pipelining is given below:
**Algorithm 2** Algo for Cleaning MessagesDefining cleaning_message(msg):  no_punc = [If the char isn’t in the string, use char for the char in the message. punctuation]  join_no_punc = ‘‘.join(no_punc)  clean_join_no_punc = [If word.lower() is not in stopwords, split Test punc removedjoin.split() word for word. words(‘english’)]  return clean_join_no_puncusing sklearn.feature_extraction.text import CountVectorizervect = CountVect(analyz = cleaning_message)reviews_countvect = vectorizer.fit_transform(reviews_df[‘verified_reviews’])

The sklearn library is used to vectorize and pipeline the dataset before feeding it to train the model. Creating a pipeline implies a series of perhaps functions or lines of code that can consume any text data and basically perform these three processes, remove punctuations and then remove stop words and then perform count vectorization.
Step 4: Build and train a Machine Learning Model to analyze a Smart IoT Device

In this step, we trained and tested the Naive Bayesian classifier model. In the previous step, we created a pipeline that can remove punctuations from text, remove stop words, and also perform count vectorization. And now we’re pretty much ready to go ahead and train our model. So the first step is to divide our data sets into training and testing. So in general when we train any machine learning model we use most of the data set perhaps around let’s say 80% of the data to train our model and then the remaining 20% we use them. And this is very important to make sure that our model is not overfitting the training data meaning the model still performs quite well even if the data has never been seen by the model before. And that’s why the actual true assessment of the model is on the testing data set which is a new, unique dataset for testing. In this step, initially, the entire dataset is divided into test and train data. Thereafter, the multinomial Naive Bayesian Classifier is implemented on the dataset. The test results are predicted in the form of a confusion matrix. Figure 6 shows the steps involved in the research implemented.

## 4. Results and Comparative Analysis

The model has been successfully trained and the performance of the model is visualized by plotting the confusion matrix as shown in Figure 7. The confusion matrix is simply a visual representation of our classifier model performance basically what we have here on the rows are our predictions. There are true classes in the confusion matrix, the true class represents the ground truth. The model predictions match the true class this means that the model is accurate. The model predictions were positive meaning that the model said based on what I’m reading right now from that customer review, it looks like the customer is happy for example, or maybe positive. If the true class was positive meaning if the actual customer was happy and left a positive review. The model predictions match what’s happening in real life and these are what we call through negatives basically in the diagonal elements here. If the model predicted negative and the true class was positive, we call that false-negative and we call that an error.

Table 4 shows the classification report for the predicted model. The classification report says that for Class-0, the precision is 0.81 and for class-1, the precision is around 0.95. The accuracy of our multinomial naive Bayesian classifier model came out to be 94%. The overall F1 score which is the harmonic mean between the precision and recall came out to be 0.99 for class-1.

For the comparative analysis, our proposed multinomial model is compared with the logistic regression also, the results obtained from logistic regression is shown in Table 5, it shows an accuracy of 93%, whereas our multinomial model obtained an accuracy of 94%.

To analyze the performance of our multinomial model, a comparative study is done with similar models in the same domain. After a comparative study as shown in Table 6, it has been observed that our model is the second-best model in the same domain. The metric used for the comparative study is the accuracy of the model.

Text data from a number of sources, such as Facebook, Twitter, and Amazon, may be analysed and information can be extracted using a process known as sentiment analysis, sometimes known as opinion mining. In order for firms to actively work on enhancing their company strategy and gaining a complete understanding of the buyer’s viewpoint on their products, it is crucial. It comprises a computer study of a person’s buying habits, followed by a search for his opinions on a company’s commercial body. An activity, a person, a blog piece, or a shopping experience may all be examples of this entity. For this study, Amazon provided a dataset that included reviews of cameras, computers, mobile phones, tablets, televisions, and video surveillance. We used machine learning techniques to categorize positive and negative evaluations after pre-treatment. Machine Learning Techniques provide the greatest results for classifying Product Reviews, according to this research [22].

There are three types of reviews: favorable, impartial, and negative. This is useful not just for customers who want to read product reviews before buying, but also for businesses who want to see how the public reacts to their items. Using the Amazon API, we [23,24,25,26,27,28] were able to retrieve Amazon reviews. Machine learning classifiers have also been trained using unigrams and weighted unigrams. The results demonstrate that machine learning methods perform well on weighted unigrams, with SVM achieving the highest level of reliability [29,30,31,32,33,34]. The formulae used in the calculation of the classification report are given below:(1)$$Accuracy=\left(TP+TN\right)\;/\;\left(TP+TN+FP+FN\right)$$
(2)$$Precision=\left(TP\right)\;/\;\left(TP+FP\right)$$
(3)$$Recall=\left(TP\right)\;/\;\left(TP+FN\right)$$
(4)$$F1-score=\left(2\;\times \;Precision\;\times \;Recall\right)\;/\;\left(Precision+Recall\right)$$
where *TP* = True Positive, *TN* = True Negative, *FP* = False Positive, *FN* = False Negative.

Numerous opinion mining approaches are used to extract the feelings hidden in the comments and reviews for a specific unlocked mobile. Additionally, a thorough analysis of sentiment classification methods is performed using the data set from mobile phone assessments [35,36,37,38,39,40,41,42]. The study’s [24] findings provide a comparison of eight distinct classifiers based on the assessment parameters of accuracy, recall, precision, and F-measure. Random Forest Classifiers are more efficient than other classifiers, however, LSTM and CNN are also more accurate.

Convolutional and max-pooling layers let the CNN model successfully retrieve higher-level data. With the help of the LSTM model, relationships between word sequences across time may be recorded. In this paper [25], we present a Hybrid CNN-LSTM Model, which combines LSTM with an extremely deep CNN model to solve the sentiment analysis problem. The proposed model additionally incorporates a rectified linear unit, dropout technology, and normalising to enhance accuracy. The proposed Hybrid CNN-LSTM Model outperforms conventional deep learning and ML methods in terms of precision, recall, f-measure, and accuracy.

## 5. Conclusions

NLP is used in the method of sentiment analysis to ascertain the attitude or sentiment of a text. A sentiment analysis algorithm may determine if a given text data is positive, negative, or neutral by deriving information from natural language and assigning it to a numerical score. There are several methods for developing or training a sentiment analysis model, and sentiment analysis is used by a variety of businesses to better understand their customers’ sentiments through reviews and social media conversations and make more timely and accurate business decisions. This study aims to offer a technique for assessing Amazon Echo customer evaluations and classifying them as good or negative. A dataset including the reviews of 3150 people was employed in this study. Initially, a word cloud of excellent and negative evaluations was produced, which provided a great deal of information from text data. After that, 80 percent of the dataset was used to build and train a machine learning model using a multinomial naive Bayesian classifier. The remaining 20% of the dataset was utilized to evaluate the model. The proposed model’s accuracy rate is 94%. The proposed model outscored three of the four models in the same area.

## Figures and Tables

**Figure 1 sensors-22-07318-f001:**
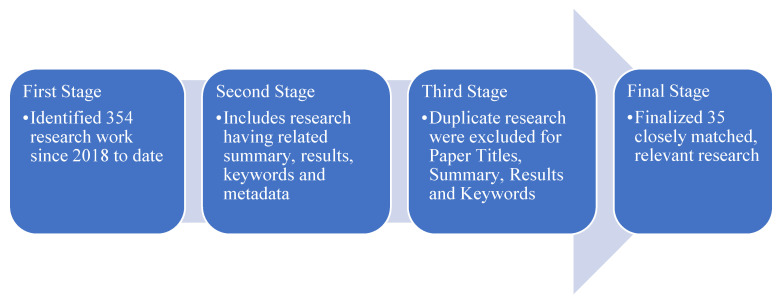
Research Selection Methodology.

**Figure 2 sensors-22-07318-f002:**
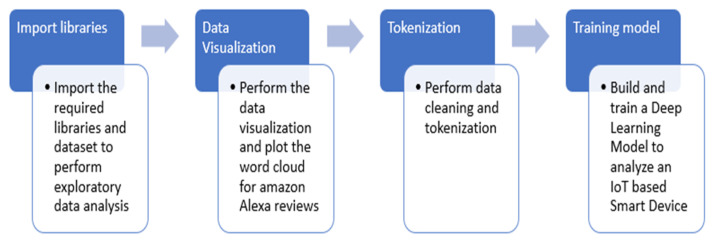
Research Methodology.

**Figure 3 sensors-22-07318-f003:**
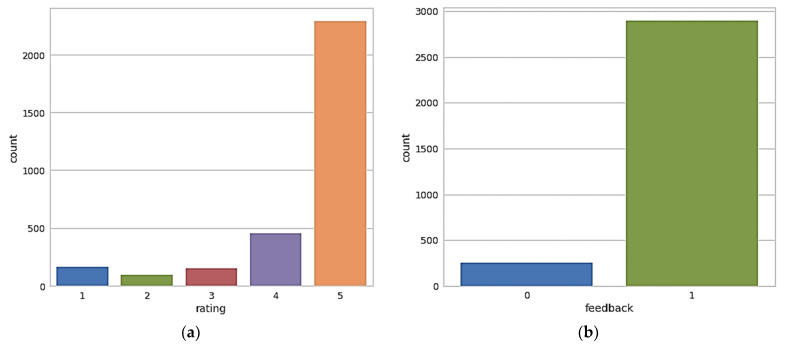
(**a**): Plot of rating counts, (**b**): Plot of feedback counts.

**Figure 4 sensors-22-07318-f004:**
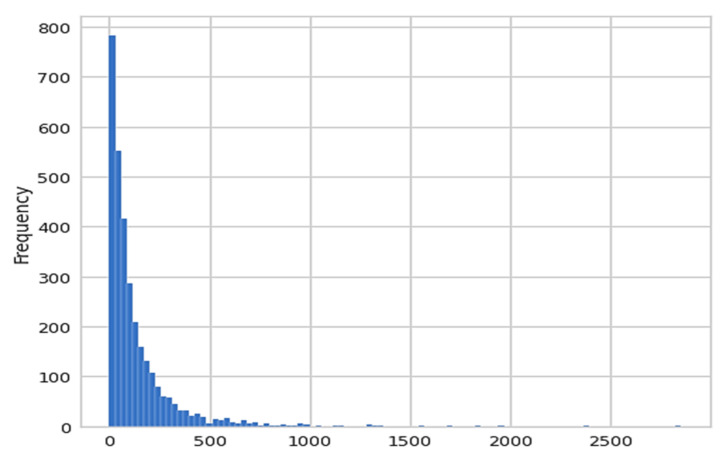
Histogram plot for the length of the review.

**Figure 5 sensors-22-07318-f005:**
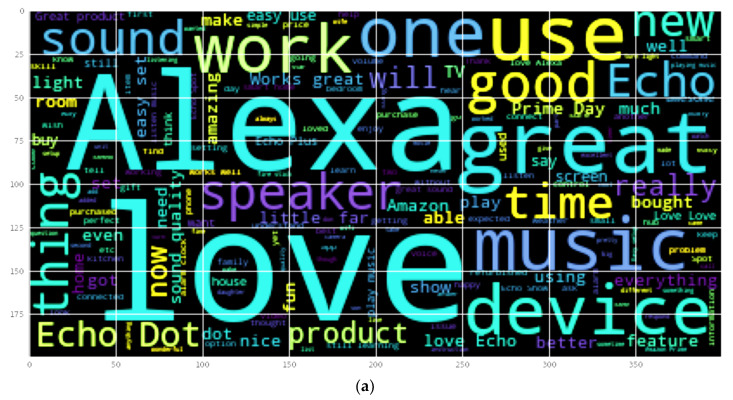
(**a**) Wordcloud of positive reviews. (**b**) Wordcloud of negative reviews.

**Figure 6 sensors-22-07318-f006:**
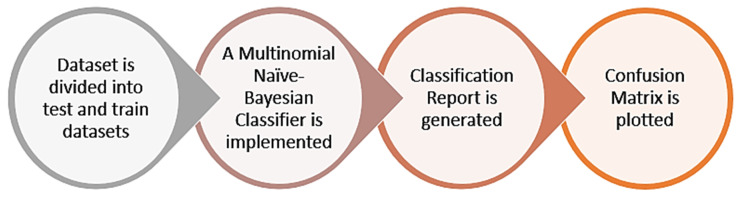
Research Implemented.

**Figure 7 sensors-22-07318-f007:**
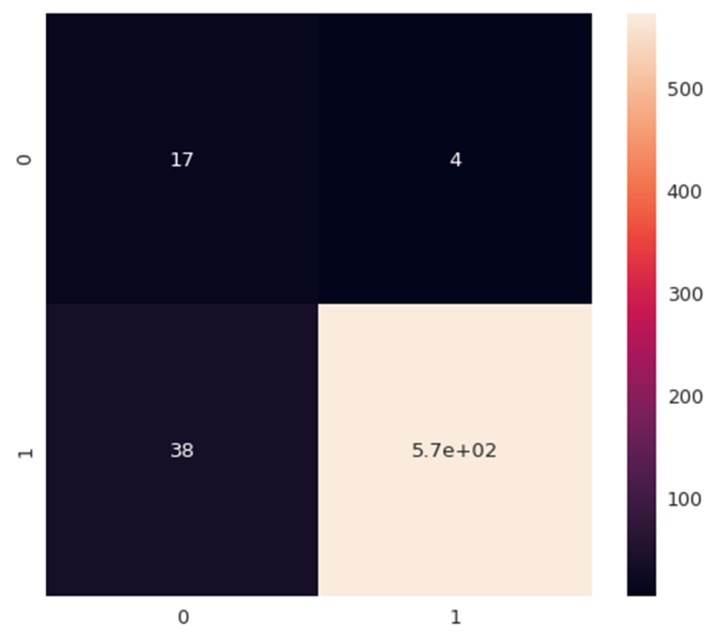
Confusion Matrix.

**Table 1 sensors-22-07318-t001:** Research Selection & Classification.

Grading Classification	Stage 1	Stage 2	Stage 3	Stage 4	Breakup
**Machine Learning**	71	45	18	7	**20.06%**
**Artificial Intelligence**	82	52	20	8	**23.16%**
**Alexa**	69	44	17	7	**19.49%**
**Voice Assistant**	63	40	16	6	**17.80%**
**IoT devices**	69	44	17	7	**19.49%**
	**354**	**227**	**88**	**35**	

**Table 2 sensors-22-07318-t002:** Dataset Information.

Sr. No.	Dataset Column	Non-Null Count	Datatype
1	rating	3150	int64
2	date	3150	Object
3	variation	3150	Object
4	verified_reviews	3150	Object
5	feedback	3150	int64

**Table 3 sensors-22-07318-t003:** Statistical Summary of the dataset.

Statistics	Rating	Feedback
count	3150.000000	3150.000000
mean	4.463175	0.918413
std	1.068506	0.273778
min	1.000000	0.000000
25%	4.000000	1.000000
50%	5.000000	1.000000
75%	5.000000	1.000000
Max	5.000000	1.000000

**Table 4 sensors-22-07318-t004:** Classification Report.

Classification	Precision	Recall	F1-Score	Support
**0**	0.77	0.46	0.57	59
**1**	0.95	0.99	0.97	571
**Accuracy**			0.94	630
**Macro average**	0.86	0.72	0.77	630
**Weighted average**	0.93	0.94	0.92	630

**Table 5 sensors-22-07318-t005:** Classification Report of Logistic Regression.

Classification	Precision	Recall	F1-Score	Support
**0**	0.85	0.29	0.43	59
**1**	0.93	0.99	0.96	571
**Accuracy**			0.93	630
**Macro average**	0.89	0.64	0.70	630
**Weighted average**	0.92	0.93	0.91	630

**Table 6 sensors-22-07318-t006:** Comparative Analysis with similar approaches.

Related Research Papers	Accuracy
[22]	98.17%
[23]	66.84%
[24]	70.55%
[25]	92.0%
Proposed Method	94.0%

## Data Availability

The study did not report any data.

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
