# Peer review of "Multinomial Naive Bayesian Classifier Framework for Systematic Analysis of Smart IoT Devices"

_sensors, 2022, doi:10.3390/s22197318_

Round 1

Reviewer 1 Report

This paper proposes a classifier model for user satisfaction analysis for smart IoT devices (e. g. Amazon echo). The main goal of the work is to understand customer sentiment based on the review provided by the users.

  1. The quality of some figures needs to be improved, for example, Fig. 2, Fig. 3, Fig. 4 ……
  2. I can't see much difference between fig 5a and 5b. Specifically, 5b does not have many glimpses of negative reviews.
  3. The proposed method in this work has less accuracy compared to [22]; why is that?
  4. You need to compare your work with other classifiers. Therefore, I would suggest reconducting the experiments and compare with different classifiers.
  5. How did you use deep learning in your proposed work?
  6. The overall organisation and presentation need to be thoroughly revised; in its present form, the experimental evaluation and implementation of the proposed method are unclear and hard to follow.

Author Response

Reviewer 1 comments

Authors response

The quality of some figures needs to be improved, for example, Fig. 2, Fig. 3, Fig. 4 …….

(1)   The quality of some figures needs to be improved, for example, Fig. 2, Fig. 3, Fig. 4 …….

Author Response: Dear Reviewer, thanks for the comments, we have improved the quality of all the figures in the paper.

Changes performed: We have re-drawn the figure 2 and qualities of rest other figures are improved.

Figure 2: Research Methodology

(2)   I can't see much difference between fig 5a and 5b. Specifically, 5b does not have many glimpses of negative reviews.

Author Response: Dear Reviewer, that you for your suggestion.

Changes performed: If both the figures are carefully observed, we will notice the negative words in figure 5b like – “refurbished”, “problem”, “disappointed”, “never”, “return”, “tried”, “nothing”, etc. The dataset used for this paper had mixture of positive and negative words. Obviously, the number of positive reviews are comparatively more than the negative reviews and the same has been highlighted in Figure 5 (a and b). Please refer to figure 3b, out of 3150 review feedback, only 250 are negative reviews

(3)   The proposed method in this work has less accuracy compared to [22]; why is that?

Author Response: Dear Reviewer, that you for the comment. In [22], the authors used a dataset that was taken from Amazon which contains reviews of Camera, Laptops, Mobile phones, tablets, TVs, video surveillance, whereas, we focused on Amazon Alexa comments only. Therefore, there is a difference in accuracy. Our model achieved an accuracy of 94% which is best as per all the research available on Amazon Alexa.

(4)   You need to compare your work with other classifiers. Therefore, I would suggest reconducting the experiments and compare with different classifiers.

Author Response: Dear Reviewer, that you for your suggestion.

Changes performed: Dear Reviewer, For the comparative analysis, our proposed multinomial model is compared with the logistic re-gression also, the results obtained from logistic regression is shown in table 5, it shows an accuracy of 93%, whereas our multinomial model obtained an accuracy of 94%.

Classification

Precision

Recall

F1-score

Support

0

0.85

0.29

0.43

59

1

0.93

0.99

0.96

571

Accuracy

0.93

630

Macro average

0.89

0.64

0.70

630

Weighted average

0.92

0.93

0.91

630

(5)   How did you use deep learning in your proposed work?

Author Response: Dear Reviewer, that you for your question.

Changes performed: Dear Reviewer, the authors have not used deep learning but have implemented the machine learning. The authors have implemented the Naïve Bayesian classifier that comes under machine learning and the

(6)   The overall organisation and presentation need to be thoroughly revised; in its present form, the experimental evaluation and implementation of the proposed method are unclear and hard to follow.

Author Response: Dear Reviewer, that you for your suggestion.

Changes performed: Dear Reviewer, as per your suggestion, the paper is now organized. In section 3, research methodology is discussed mentioning the process followed by the authors in the implementation. Followed by results and comparative analysis in section 4. In this section the results obtained are highlighted and the comparative analysis with existing approaches/work is done. At last, in section 5, the paper is concluded.

Reviewer 2 Report

The paper's main objective is to analyze amazon reviews for the echo product and predict customer sentiment. The paper is well-written in general, but I have some suggestions.

1) What is the research string used in Related work?

2) The authors should insert a comparative table in the section Related work to clarify their main contribution.

3) Several figures (for example Figures 5a and 5b) are too big. Please adjust their size.

4) Why only the Naive Bayesian classifier model is trained? Why the authors do not compare other classifiers?

5) Table 4: Th authors should explain F1-score 0.45.

6) The authors should explain in detail the Machine learning classifiers trained using unigrams and weighted unigrams. They mention SVM, but they do not explain the process.

7) What is the relation and extension of their paper when we compare it to references [23] [24] and [25]? The last three paragraphs (before the Conclusion section) are not clear.

8) Table 5 should be explained in a better way. Which methods are used in the paper compared with their work?

Author Response

Reviewer 2 comments

Authors response

1) What is the research string used in Related work?

Author Response: Dear Reviewer, thanks for the query. The research string used in the related work is highlighted in Table 1: Research Selection & Classification. For finding out the related work, proper research selection and classification is done as per the statistics mentioned in table 1. Most relevant papers are then referred and thus added in the literature part of this paper.

Changes performed: Dear Reviewer, we have highlighted the table 1 for your reference. Thanks.

2) The authors should insert a comparative table in the section Related work to clarify their main contribution

Author Response: Dear Reviewer, thankyou for your suggestions. The authors have added table 5 for comparative analysis of our approach with the results obtained by logistic regression.

Changes performed: Dear Reviewer, as per your suggestion, we have updated the research paper with table 5.

3) Several figures (for example Figures 5a and 5b) are too big. Please adjust their size.

Author Response: Dear Reviewer, thankyou for your suggestion. We can surely reduce the size of the figures but then the figure content will not be properly visible to the readers, that’s why the figures are deliberately kept large in size.

Changes performed: Dear Reviewer, we may reduce the dimension of the figures (if required).

4) Why only the Naive Bayesian classifier model is trained? Why the authors do not compare other classifiers.

Author Response: Dear Reviewer, thankyou for your suggestions. As per your suggestion, we have added one more comparison table (table 5) that is related to the results obtained from the logistic regression.

Changes performed: Dear Reviewer, we have updated the research paper as per your suggestion and the same has been highlighted in the manuscript.

5) Table 4: The authors should explain F1-score 0.45.

Author Response: Dear Reviewer, thankyou for your query. The author has mentioned the calculation of F1-score in equation no. 4. F1-score is calculated as .

Changes performed: Dear Reviewer, we have highlighted the equation 4 in the paper for the F1-score calculation for your reference.

6) The authors should explain in detail the Machine learning classifiers trained using unigrams and weighted unigrams. They mention SVM, but they do not explain the process.

Author Response: Dear Reviewer, thankyou for your query. In this paper, we have not implemented SVM, but, it was used in paper [23]. Therefore, over there the authors have mentioned about SVM.

7) What is the relation and extension of their paper when we compare it to references [23] [24] and [25]? The last three paragraphs (before the Conclusion section) are not clear.

Author Response: Dear Reviewer, thankyou for your query. These paragraphs are added just to do the comparison with our paper. If required, these paragraphs can be removed from the results section and can be moved to related work section (section 2).

8) Table 5 should be explained in a better way. Which methods are used in the paper compared with their work?

Author Response: Dear Reviewer, thankyou for your query. Table 5 (now table 6) is used for the comparative analysis of related work. The accuracy of various related research works is compared, and it has been observed that our model is the second-best model in the same domain.

Round 2

Reviewer 1 Report

Thanks for addressing suggested comments. However, I did not find the abstract in the paper.

Author Response

Comments and Suggestions for Authors Response
Thanks for addressing suggested comments. However, I did not find the abstract in the paper. Thankyou so much for your comments sir , we have added the abstract in the updated manuscript .

Reviewer 2 Report

The authors should revise the entire paper in order to improve a few English mistakes and typos.

Author Response

Comments and Suggestions for Authors Response
The authors should revise the entire paper in order to improve a few English mistakes and typos. thankyou for your suggestion sir , we have improved the english  and removed all the typos .